# Synthesis and Characterization of Mannosylated Formulations to Deliver a Minicircle DNA Vaccine

**DOI:** 10.3390/pharmaceutics13050673

**Published:** 2021-05-07

**Authors:** Ana Sofia Serra, Dalinda Eusébio, Ana Raquel Neves, Tânia Albuquerque, Himanshu Bhatt, Swati Biswas, Diana Costa, Ângela Sousa

**Affiliations:** 1CICS-UBI—Health Sciences Research Centre, Universidade da Beira Interior, Avenida Infante D. Henrique, 6200-506 Covilhã, Portugal; anasofiamms@gmail.com (A.S.S.); dalinda-21@hotmail.com (D.E.); ana_raquel_bastos_neves@hotmail.com (A.R.N.); t_albuquerque@live.com.pt (T.A.); 2Department of Pharmacy, Nanomedicine Research Laboratory, Birla Institute of Technology & Science-Pilani, Hyderabad Campus, Jawahar Nagar, Medchal, Hyderabad, Telangana 500078, India; bhattrx@yahoo.com (H.B.); swati.biswas@hyderabad.bits-pilani.ac.in (S.B.)

**Keywords:** HPV infection, mannose ligands, minicircle DNA vaccine, polyethyleneimine, R8 peptide

## Abstract

DNA vaccines still represent an emergent area of research, giving rise to continuous progress towards several biomedicine demands. The formulation of delivery systems to specifically target mannose receptors, which are overexpressed on antigen presenting cells (APCs), is considered a suitable strategy to improve the DNA vaccine immunogenicity. The present study developed binary and ternary carriers, based on polyethylenimine (PEI), octa-arginine peptide (R8), and mannose ligands, to specifically deliver a minicircle DNA (mcDNA) vaccine to APCs. Systems were prepared at various nitrogen to phosphate group (N/P) ratios and characterized in terms of their morphology, size, surface charge, and complexation capacity. In vitro studies were conducted to assess the biocompatibility, cell internalization ability, and gene expression of formulated carriers. The high charge density and condensing capacity of both PEI and R8 enhance the interaction with the mcDNA, leading to the formation of smaller particles. The addition of PEI polymer to the R8-mannose/mcDNA binary system reduces the size and increases the zeta potential and system stability. Confocal microscopy studies confirmed intracellular localization of targeting systems, resulting in sustained mcDNA uptake. Furthermore, the efficiency of in vitro transfection can be influenced by the presence of R8-mannose, with great implications for gene expression. R8-mannose/PEI/mcDNA ternary systems can be considered valuable tools to instigate further research, aiming for advances in the DNA vaccine field.

## 1. Introduction

Cancer is a disease that continues to proliferate worldwide and is a major cause of mortality, however the probability of finding a cure is increasing. Cervical cancer is the most relevant disease associated with human papillomavirus (HPV) infection, especially when it is not detected early, evolving to invasive forms [1]. The overexpression of HPV E6 and E7 oncoproteins interferes with cell cycle regulation and proliferation through the impairment of p53 and pRb tumor suppressor proteins, respectively [2]. Although prophylactic HPV vaccination is currently the best strategy for preventing cervical cancer, this type of vaccine induces only humoral immunity (producing neutralizing antibodies) to prevent future infections, however it is not effective in treating pre-existing infections. To fill this gap, several types of therapeutic vaccines against HPV are being studied [3]. In this context, DNA vaccines have particular importance due to their ability to generate cellular and humoral immune responses, based on the use of genetic material sequences from the pathogen that is intended to be fought [4]. Among the different DNA vaccines, minicircle DNA (mcDNA) is an innovative and promising DNA molecule. The absence of prokaryotic sequences on the mcDNA overcome some key limitations of the traditional plasmid DNA [5]. E6 and E7 viral oncoproteins are suitable targets for the formulation of therapeutic vaccines, since they are overexpressed in HPV-infected cells at early stages [1].

The success of DNA vaccines is strongly dependent on the development of convenient and efficient gene delivery systems. They should be able to condense and protect the DNA, binding to the membrane and internalizing in eukaryotic cells, overcoming all intracellular and extracellular obstacles [6]. The intracellular delivery of nucleic acids is thought to result from the electrostatic interactions between the positive charges of the delivery system and the negative charges of the cellular membrane. Higher surface charges mean stronger DNA loading capability, which favors effective cell uptake and gene transfection [7]. Viral vectors offer high transfection efficiency for eukaryotic cells. However, they present significant disadvantages, such as antigenicity, potential oncogenic effects, possible virus recombination, or difficulty for large-scale production and storage. To overcome these limitations, several attempts have been made to develop non-viral gene delivery systems based on liposomes, synthetic or natural cationic polymers, and cell-penetrating peptides (CPPs), among others [8].

Among non-viral delivery methods, cell-penetrating peptides (CPPs) have received particular attention due to their ability to enter cells in a non-invasive manner, maintaining the integrity of cell membranes, and they are considered highly efficient and safe [9]. These peptides are generally short at up to 30 amino acids and can be separated into two main groups: arginine-rich and amphipathic peptides [10]. Although the delivery properties of CPPs are not questionable, the mechanism of cell uptake and endosomal escape is still controversial [10]. Different studies are currently underway to better understand this mechanism. For instance, the translocation efficiency and cell localization of several arginine-rich peptides with various chain lengths have been analyzed. In particular, the octa-arginine (R8) peptide has shown very satisfactory results in terms of cell internalization and nucleus accumulation [11,12,13,14,15]. In addition, CPPs can also be used to enhance nanosystem activity and biocompatibility [16].

Cationic polymers have been extensively studied due to their ability to condense DNA into polyplexes and facilitate gene delivery [8,17,18]. Polyethylenimine (PEI) is the most used synthetic cationic polymer, existing in various molecular weights and linear or branch structures. For instance, branched PEI with higher molecular weights shows greater ability to condense nucleic acids than linear PEI [19]. The abundance of polyamines favors the formulation of PEI/DNA nanocarriers with high surface charge, which improves the transfection efficiency but also increases the toxicity. On the other hand, PEI-based DNA carriers with low-molecular-weight are non-toxic but exhibit low transfection efficiency. As such, PEI conjugation with other molecules is a possible approach to reduce toxicity [7].

Antigens encoded in DNA vaccines need to be processed by antigen-presenting cells (APCs) to activate both humoral and cellular immune responses [20]. To increase the efficiency of DNA vaccines in terms of immunogenicity, several strategies have focused on targeting APCs [21]. Dendritic cells (DCs) and macrophages represent the crucial APCs for efficient activation of T-cells and B-cells and are concentrated in areas of potential antigen entry, especially around the epithelial and mucous surfaces [22]. Therefore, the development of new and improved formulations containing specific ligands to target and deliver DNA vaccines into APCs has been explored [21]. Mannose ligands are often used to bind mannose receptors that are highly expressed on surfaces of DC and macrophages. Some studies have evidenced the entrance of mannosylated formulations into Raw 264.7 cells through receptor-mediated endocytosis [23,24]. In addition, the three-dimensional conformation and multimerization pattern of each mannose receptor determine its ability to recognize multiple ligands and regulate immune responses [25].

In this work, binary and ternary delivery systems based on R8 peptide functionalized with mannose ligands with and without the PEI polymer were designed to specifically deliver the mcDNA vaccine to macrophages. R8-mannose/mcDNA and R8-mannose/PEI/mcDNA formulations were explored, optimized, and characterized in terms of their surface morphology, size, surface charge, complexation capacity, and stability. Transfection studies were conducted to assess the biocompatibility, cell internalization, and expression of the genes encoded by the mcDNA vaccine carried by the developed systems. The presented report provides relevant data on the conception, characterization, and transfection efficiency of non-viral delivery systems and is a great contribution to progress in DNA vaccines.

## 2. Materials and Methods

### 2.1. Materials

α-d-Mannopyranosylphenyl isothiocyanate (MPITC) and commercial branched PEI with average Mw 25 kDa and GRS Taq DNA polymerase were purchased from Sigma Aldrich Chemicals (St. Louis, MO, USA). R8 peptide was custom synthesized from GCC Biotech Pvt. Ltd. (West Bengal, India). The TripleXtractor used in RNA extraction was obtained from GRISP (Porto, Portugal). DMEM-F12 and DMEM-HG were purchased from GIBCO (Waltham, MA, USA). Sodium bicarbonate was obtained from MP Biomedicals (Santa Ana, CA, USA). DAPI was obtained from Invitrogen (Carlsbad, CA, USA). FITC was obtained from Alfa Aesar (Lanchashire, UK). Agarose and GreenSafe were obtained from NZYtech (Lisbon, Portugal).

### 2.2. Methods

#### 2.2.1. Amplification and Purification of mcDNA Vector

To eradicate the oncogenic potential of HPV E7, the HPV E7 gene was modified by NZYTech (Lisbon, Portugal) to include three point mutations preventing the interaction with pRB but maintaining the normal structure of the E7 protein (more information in Appendix A). The mutated E7 gene was cloned into the pMC.CMV-MCS-EF1-GFP-SV40PolyA parental plasmid (PP) vector (next to CMV7 promoter), which was amplified in the ZYCY10P3S2T *Escherichia coli* host strain following production under the induction conditions described by our research group [26]. Briefly, the bacterial cultures were performed in 1 L Erlenmeyer flasks containing 250 mL of Terrific Broth medium (20 g/L of tryptone; 24 g/L of yeast extract; 4 mL/L of glycerol; 0.017 M KH_2_PO_4_, and 0.072 M K_2_HPO_4_, pH 7.0) in an orbital shaker at 42 °C and 250 rpm. For the mcDNA-E7 vector production from its PP precursor, an induction mix containing L-arabinose 0.01% (*w*/*v*) was added. The recombination process was carried out for 2 h at 32 °C. When induction was completed, cells were recovered by centrifugation and stored at −20 °C. To obtain the mcDNA-E7 vector, a modified alkaline lysis method was performed as described by Diogo et al. [27]. To isolate the mcDNA vector, resulting supernatants were loaded directly onto a Sephacryl SF-1000 instrument, as previously described by our research group [28]. An AKTA Pure system (GE Healthcare, Buckinghamshire, UK) with UNICORNTM 6.3 software (GE Healthcare, Buckinghamshire, UK) was used to perform all chromatographic runs. The resultant chromatographic fractions were desalted and concentrated using Vivaspin^®^ 6 centrifugal concentrators (Vivaproducts, Littleton, MA, USA) and analyzed by agarose gel electrophoresis.

#### 2.2.2. Agarose Gel Electrophoresis

The 0.8% or 1% (*w*/*v*) agarose gel (0.4 g or 0.5 g of agarose) was prepared for 50 mL of 1× TAE buffer (40 mM Tris base, 20 mM acetic acid, 1 mM EDTA at pH 8.0) and stained with 0.6 μL of GreenSafe. Electrophoresis was performed for 40 min at 120 V and the gel was analyzed using ultraviolet (UV) light through the Uvitec Fire-Reader system (UVITEC, UK).

#### 2.2.3. Synthesis of α-d-Mannopyranosylphenyl Isothiocyanate-Octa-Arginine Conjugate

Mannose was conjugated to R8 via the reaction between isothiocyanate and the amine groups (Figure 1). Briefly, α-d-mannopyranosylphenyl isothiocyanate (MPITC) was dissolved in methanol and added dropwise into the methanolic solution of R8 at the MPITC/R8 mole ratio of 1.2:1 under stirring for 24 h. The methanol was evaporated under vacuum using a rotary evaporator.

#### 2.2.4. Characterization of Synthesized MPITC-R8 Conjugate

The characterization of the synthesized MPITC-R8 conjugate was obtained using 1H NMR of the synthesized conjugate and size exclusion chromatography (SEC). 1H NMR was performed by dissolving the compound (10 mg) in deuterated methanol (CD3OD) (1 mL) using a Bruker spectrometer (300 MHz, Bruker, Billerica, MA, USA) operating at 300 mega Hz at 25 °C. The SEC was performed to estimate the molecular weight of the synthesized MPITC-R8 conjugate. The samples were eluted through an Ultrahydrogel™ linear SEC column (7.8 mm × 300 mm) in a gel permeation chromatography (GPC) system (Waters Corporation, St. Louis, MO, USA). Milli-Q water was used as a mobile phase with a flow rate of 0.7 mL/min. The SEC standards were run before analyzing the conjugates.

#### 2.2.5. Preparation of R8-Mannose/mcDNA and R8-Mannose/PEI/mcDNA Complexes

R8-mannose lyophilized powder was suspended in ultrapure water and aliquots of 0.5 mg/mL were prepared. PEI stock solutions were prepared in sodium acetate buffer (0.1 mM sodium acetate/0.1 M acetic acid, pH 4.5). R8-mannose/mcDNA-E7 binary and R8-mannose/PEI/mcDNA-E7 ternary systems were prepared, characterized, and evaluated to determine their potential as delivery systems. Formulation of R8-mannose/mcDNA-E7 binary particles was achieved by adding variable concentrations of peptide solution to a fixed amount of mcDNA (1 μg) under vortexing for 1 min. To formulate R8-manose/PEI/mcDNA-E7 ternary systems, different amounts of PEI were added to a fixed volume of mcDNA and then variable concentrations of R8-mannose were added to PEI/mcDNA-E7 particles under vortexing for 1 min. All systems were left for 30 min at room temperature to allow particle formation and then centrifuged at 10,000 rpm for 20 min at 4 °C.

Both systems were prepared at various N/P ratios, considering the molar ratio of positively charged amine groups from R8 and PEI (N) to negatively charged phosphates in the DNA backbone (P). The electrophoretic mobility of the supernatants from all nanoparticle formulations was evaluated by agarose gel electrophoresis of 1% to ensure the entire mcDNA amount was complexed (Appendix A).

#### 2.2.6. Characterization of Systems

Fourier transform infrared spectroscopy (FTIR) was applied to investigate the functional groups on the particle surfaces. The pellet recovered from different formulations was suspended in 10 μL of ultrapure water. The spectra were acquired using a Nicolet iS10 FTIR spectrophotometer (Thermo Scientific, Waltham, MA, USA) with an average of 120 scans, a spectral width ranging from 4000 and 600 cm^−1^, and a spectral resolution of 32 cm^−1^. The spectra of isolated R8 and mcDNA samples were acquired for comparative analysis.

Scanning electron microscopy (SEM) was used to obtain information concerning the morphology of both systems. Different formulations were centrifuged and the pellet was recovered and suspended in an aqueous solution containing 40 μL of tungsten. The solution was placed in a round-shaped cover slip and dried overnight at room temperature. The samples were sputter-coated with gold using an Emitech K550 (London, England) sputter coater. A Hitachi S-2700 (Tokyo, Japan) scanning electron microscope with an accelerating voltage of 20 kV at various magnifications was used to analyze the morphologies of binary and ternary systems.

The average size and zeta potential of the particles were determined via dynamic light scattering (DLS) at 25 °C using a Zetasizer nano ZS device (Malvern Instruments, Worcestershire, UK). DLS techniques were performed with a He-Ne laser at 633 nm with non-invasive backscatter (NIBS) to assess systems size and with electrophoretic light scattering optics using a M3-PALS laser (phase analysis light Scattering) for charge characterization. All experiments were performed in triplicate and were analyzed using Malvern zetasizer software v 6.34 (Malvern Instruments, Worcestershire, UK). The pellet containing the particles was suspended in 5% glucose with 1 mM NaCl.

#### 2.2.7. Stability Assays

R8-mannose/mcDNA-E7 and R8-mannose/PEI/mcDNA-E7 systems were incubated for different time periods (0, 1, and 4 h) with 25 μL of DMEM medium supplemented with 10% FBS and 25 μL of trypsin solution at 37 °C. The release and mcDNA degradation were monitored by 1% agarose gel electrophoresis.

#### 2.2.8. In Vitro Transfection

Cell culture experiments were performed using human fibroblast cells (ATCC^®^ PCS-201-012™) and Raw 264.7 cells (murine macrophage cells, ATCC^®^ TIB-71™). Human fibroblast cells were grown with Dulbecco’s modified Eagle’s medium with Ham’s F-12 Nutrient Mixture (DMEM-F12) supplemented with 10% heat-inactivated fetal bovine serum, 2.438 g/L sodium bicarbonate, and 1% (*v*/*v*) of a mixture of antibiotics composed of penicillin (100 µg/mL) and streptomycin (100 µg/mL). Raw 264.7 cells were grown with Dulbecco´s modified Eagle´s medium with High Glucose (DMEM-HG) supplemented with 10% non-inactivated fetal bovine serum, 1.5 g/L sodium bicarbonate, and with 1% (*v*/*v*) of a mixture of antibiotics composed of penicillin (100 µg/mL) and streptomycin (100 µg/mL). The cellular growth was promoted at 37 °C in a humidified atmosphere containing 5% CO_2_. For transfection studies, human fibroblast cells and Raw 264.7 cells were seeded in 12-well plates at densities of 2.5 × 10^5^ cells/well and 2 × 10^5^ cells/well, respectively, in 1 mL complete medium. After 24 h and before transfection occurred, the medium was replaced by a medium without FBS and antibiotic supplementation (incomplete medium) in order to promote transfection. At confluency (50–60%), the medium was removed and the cells were transfected with different particles dissolved in incomplete medium. For human fibroblast transfection, 7.5 µg of encapsulated mcDNA from each system was added per well. Raw 264.7 cells were transfected by adding 6 µg of encapsulated mcDNA from each system per well. After 6 h of transfection, the incomplete medium was replaced by complete medium.

#### 2.2.9. Biocompatibility Study

A resazurin assay was used in order to evaluate the systems’ biocompatibility. Human fibroblasts and Raw 264.7 cells were seeded in 96-well plates at densities of 1 × 10^4^ cells/well and 0.8 × 10^4^ cells/well, respectively. For human fibroblasts, 0.3 µg of encapsulated mcDNA from each system was added per well. Raw 264.7 cells were transfected by adding 0.24 µg of encapsulated mcDNA from each system per well. After 24 and 48 h of transfection, the culture medium was discarded and 100 µL of fresh complete medium and 20 µL of resazurin 0.1% (*w*/*v*) were added to each well and incubated over four hours in the dark at 37 °C in a humidified atmosphere of 5% CO_2_. After incubation, the fluorescence was measured in a spectrofluorometer (SpectraMAX^®^ GeminiTM EM, Molecular Devices, San Jose, CA, USA) at an excitation wavelength of 544 nm and emission wavelength of 590 nm to analyze the resorufin fluorescence produced.

#### 2.2.10. FITC Plasmid Labeling

Minicircle DNA was stained with FITC by assembling 16.3 μL of mcDNA, 2 μL of FITC, and 66.7 μL of labeling buffer. Samples were placed under constant stirring at room temperature for 4 h and protected from light. One volume of 3 M NaCl (85 μL) and 2.5 volumes of 100% ethanol (212.5 μL) were added. Samples with stained mcDNA were incubated at −20 °C overnight. Subsequently, samples were centrifuged at 4 °C for 30 min and the pellet was washed with 75% ethanol.

#### 2.2.11. Cellular Uptake and Internalization

Confocal fluorescence microscopy was used to investigate the cellular uptake and internalization of carriers. Raw 264.7 cells were grown in an 8-well μ-slide (Ibidi, Martinsried, Germany) until 50–60% confluence was achieved. Nuclei were stained by incubating the cells with DAPI. FITC-labeled mcDNA was encapsulated into R8-mannose/PEI particles and real live transfection was visualized using an LSM 710 confocal laser scanning microscope (Carl Zeiss, Oberkochen, Germany) under 63× magnification and analyzed with the Zeiss LSM 710 laser scanning confocal microscope (Carl Zeiss SMT, Inc., Oberkochen, Germany). During the experiment, Raw 264.7 cells were maintained at 37 °C with 5% CO_2_.

#### 2.2.12. Reverse Transcription Polymerase Chain Reaction (RT-PCR)

RT-PCR was used to detect the E7 mRNA transcripts resulting from the E7 gene transcription encoded in the PEI/mcDNA and R8-mannose/PEI/mcDNA system vector. After 24 h of transfection, cells were lysed through the addition of TripleXtractor and incubated at room temperature for 5 min. Subsequently, 50 μL of chloroform was added and stirred to allow the separation of different biomolecules in different phases, then incubated at room temperature for 10 min. Samples were then centrifuged at 12,000× *g* for 15 min at 4 °C to obtain the separation of the aqueous phase containing RNA and the interphase and lower organic phase containing DNA and proteins. The aqueous phase was carefully recovered and 125 μL of ice-cold isopropanol was added to precipitate the RNA. Samples were centrifuged again at 12,000× *g* for 15 min at 4 °C and the pellet was washed in DEPC water with 125 μL of 75% ethanol to eliminate organic compounds. A new centrifugation was carried out at 12,000× *g* for 5 min at 4 °C and the RNA pellet was resuspended in 20 μL of DEPC. To confirm the success of RNA extraction, electrophoresis was performed on 1% agarose gel and the samples were quantified on a NanoPhotometer™. The cDNA synthesis was performed by using Xpert cDNA Synthesis (GRiSP-Research Solutions, Porto, Portugal), following the manufacturer’s protocol. PCR amplification was performed by adding in each PCR reaction 8.25 μL of RNase-free water, 0.40 μL of forward primer (5′−AAT CTA GAA TGC CTG ATA CAC CTA C -3′) and reverse primer (5′ -ATG GAT CCT TAT GGT TTC TGA GAA CAG A -3′), 0.7 μL of MgCl2, 0.25 μL of dNTPs, 1.25 μL of PCR buffer, 0.25 μL of GRS Taq, and 1 μL of cDNA. The samples were homogenized and a mini-spin was performed. Samples were then placed in a T100™ Thermal Cycler (Bio-Rad Laboratories, Inc, Hercules, California, USA) with the following sequence: 95 °C for 5 min, 26 cycles of 30 s at 95 °C, 30 s at 60 °C, 1 min at 72 °C, and finally 10 min at 72 °C. PCR products were analyzed by electrophoresis on an agarose gel and were visualized in a UVItec Gel documentation system under UV light (UVItec Limited, Cambridge, UK).

#### 2.2.13. Reverse Transcription Quantitative Real-Time PCR (RT-qPCR)

To quantitatively analyze the levels of transcripts, RT-qPCR was performed. The mix for a reaction with primers designed for the transcript of the E7 gene was prepared with 10 μL of SYBR ™ Green Master Mix, 0.64 μL FW primer, 0.64 μL RV primer, 7.72 μL of sterile H_2_O, and 1 μL of cDNA, resulting in a volume of 20 μL per reaction. The mix for reaction with the primer pair of the GAPDH housekeeping gene transcript (FW: 5′- ATG GGG AAG GTG AAG GTC G -3′; RV: 5′- GGG GTC ATT GAT GGC AAC AAT A -3′) was prepared with 10 μL of NZY qPCR Green Master Mix (2x), 1.2 μL FW primer, 1.2 μL RV primer, 7.5 μL of sterile H_2_O, and 1 μL of cDNA, resulting in a volume of 20 μL. The reaction mixtures were placed in a Real-Time CFX ConnectTM system (BioRad, Hercules, CA, USA) programmed with the following sequence of incubations: 10 min at 95 °C, 40 cycles of 15 s at 95 °C, 30 s at 60 °C.

#### 2.2.14. Statistical Analysis

Each experience was performed at least three times. Data are expressed as means ± standard error (S.D.). The statistical analyses performed were one-way and two-way analyses of variance (ANOVA), followed by Tukey’s test. Data analysis was performed in GraphPad Prisma 6 software. Here, *p*-values below 0.05 were considered statistically significant; * *p* < 0.05; ** *p* < 0.01; *** *p* < 0.001; **** *p* < 0.0001.

## 3. Results and Discussion

### 3.1. Purification of mcDNA Vector

Size exclusion chromatography was used to perform the purification of the mcDNA vector following the conditions previously described [28]. As expected in SEC, molecules such as genomic DNA and plasmid DNA elute quickly, whereas molecules such as RNA take a longer route through the pores of the matrix and are retarded in the chromatographic column. Fractions were selected, concentrated, and desalted with Vivaspin concentrators and analyzed using 0.8% agarose gel electrophoresis. The results showed that genomic DNA eluted mostly in the first peak, PP eluted mostly in the second peak, then mcDNA molecules eluted mostly in the third peak (see details in Appendix A). The purified mcDNA vector (fractions from 10 to 15) was applied in in vitro transfection studies to verify its performance.

### 3.2. Synthesis and Characterization of MPITC-R8 Conjugate

After the procedure to synthesize the MPITC-R8 conjugate, the final product was analyzed by NMR (Figure 2). The chemical shifts at 7.3 and 1.5–2.0 in the 1NMR spectrum represented the aromatic protons present in MPITC and the methylene protons present in R8, respectively. The size exclusion chromatogram is shown in Appendix A. The relative molecular weight of MPITC-R8 is 1622 Da (Appendix A). The ratio of observed to theoretical molecular weights (0.987) indicated successful conjugation of MPITC and R8 at 1:1.

### 3.3. The Properties of R8-Mannose/mcDNA and R8-Mannose/PEI/mcDNA Complexes

The R8 peptide is a biocompatible and cationic CPP known to promote interaction and access to the inside of cells [29]. Its ability to bind, interact, and consequently condense DNA has been explored via the functionalization of systems for the delivery of genetic materials to eukaryotic cells [30,31]. Additionally, the PEI polymer has been widely applied in the formation of DNA-based systems, since it strongly interacts with DNA, condensing it and showing the capacity to efficiently deliver DNA both in vitro and in vivo [32]. PEI is a synthetic polymer that is highly soluble in water, positively charged, and whose cationic amines reduce the negative charge of DNA after complexation, causing its condensation [33]. A wide variety of ligands have also been explored to create functionalized and targeted delivery vehicles. The effectiveness of mannosylated devices is related to their ability to target mannose receptors, which are highly expressed on DCs and macrophages. Following this knowledge, the present study investigated the conception of mcDNA binary and ternary delivery systems, both of which were functionalized with mannose to enhance the interaction of formulations with macrophages.

Fourier transform infrared spectroscopy (FTIR) was used to evaluate interactions between components of each system, as well as for the presence of mannose. Figure 3 shows the FTIR spectra from different components and systems.

The spectrum of mcDNA (Figure 3A) presents peaks in the region of 1700–1500 cm^−1^ corresponding to the nitrogen bases, while the absorption peak seen at 1061 cm^−1^ is known to be related to ribose vibration (C-C sugar) [34]. The spectrum corresponding to the R8-mannose conjugate (Figure 3B) shows absorption peaks characteristic of the octa-arginine peptide already identified in other studies. The prominent peak at 1678 cm^−1^ is attributed to the elongation of guanidine N = C and carbonyl C = O and the peaks at 1199 cm^−1^ and 1157 cm^−1^ are attributed to the elongation of C (O) -O and N–C, respectively [35]. Peaks in the region of 1100 and 1000 cm^−1^ attributed to the C-O vibration of the mannose suggest its presence on the system’s surface [36,37]. The spectrum of the R8-mannose/mcDNA system (Figure 3C) suggests the existence of R8-mannose and mcDNA. The presence of R8 is evidenced by its characteristic absorption peaks, which suffered displacement due to the complexation process. It is possible to identify prominent peaks at 1642 cm^−1^, 1278 cm^−1^, and 1129 cm^−1^. The spectrum of R8-mannose/PEI/mcDNA (Figure 3D) was analyzed in order to verify whether the addition of PEI would influence the surface compositions of the systems, confirming the presence of mannose. Thus, PEI polymer shows its most important peaks at approximately 1497 cm^−1^ and 1793 cm^−1^, representing the N-H bonds that occur from encapsulation with the other components [38]. The absorption peak at 1055 cm^−1^ confirmed the presence of mannose. The absorbance peak at 970 cm^−1^ can be attributed to mcDNA. The comparison between the FTIR spectra before and after PEI conjugation confirmed its successful complexation to the R8-mannose/mcDNA system, as the amide group peaks of R8, mannose, and mcDNA were found.

Scanning electron microscopy was applied to identify the morphologies of the systems under study. Figure 4 shows images of these carriers prepared at several N/P ratios.

Both systems exhibit an oval or spherical shape, which makes them suitable for a process of cellular internalization. Previous studies have shown that spherical shaped particles benefit cell uptake and transfection efficiency [39,40]. DLS experiments were performed to obtain information regarding the mean sizes and surface charges of binary and ternary systems formulated at different N/P ratios. The respective results are presented in Figure 5.

Figure 5 shows that all particles are below 500 nm. Additionally, and for all formulations, this parameter strongly varies with the N/P ratio; it decreases with increments of the N/P ratio. The size values obtained for particles of R8-mannose/PEI/mcDNA-E7 showed that by maintaining the N/P ratios for PEI (5 and 10) and changing only the N/P ratios for R8 (1, 1.5, and 2), the particle size decreases, with the lowest value corresponding to the highest N/P ratio. The same kind of observation is valid for the situation where the R8 N/P ratios were kept constant and the amine charges from PEI were varied, e.g., when PEI N/P ratios of 5 and 10 were considered. Moreover, systems prepared at an R8 N/P ratio of 2 exhibit a significantly smaller size (234.56 nm) than for a PEI N/P ratio of 10. Due to its primary, secondary, and tertiary amines, PEI shows higher positive charge at physiological pH compared to R8 peptide. PEI also exhibits impressive endosomolytic activity since it can change its ionization degree with its pH. Additionally, the increasing molecular weight of PEI (25 kDa) and the increase of its N/P ratio will resulted in the formation of more stable and smaller systems. This can, however, induce higher cytotoxicity [41]. Nevertheless, the results obtained confirm that the increase of the amine group content will result in a greater degree of mcDNA condensation, resulting in smaller complexes [32]. The high charge density and the great condensing capacity of both PEI and R8 enhance the strong interaction with the mcDNA, leading to the formation of smaller particles.

In fact, an increase in the molecular weight of the PEI results in a decrease in the complex size and an increase in surface charge. However, for both linear and branched PEI structures, the N/P ratio is the determining parameter, since its variation can influence not only the size of the complexes but also the zeta potential they display. As the N/P ratio increases, the size of the complexes is reduced. Similarly, the effect of the N/P ratio on the surface charges is more pronounced as the N/P ratio increases [19].

Regarding the zeta potential values, it was found that R8-mannose/mcDNA-E7 system presents certain variations for the N/P ratios tested. With the N/P ratio increase, the zeta potential values become more positive. The N/P ratio of 1:1 shows a negative value (−0.67 mV), while systems with the highest tested ratio present positive surface charges (varying between +1.78 and +6.11 mV). These results indicate that the negative mcDNA charges can be neutralized by increasing the N/P ratio. Regarding the zeta potential values achieved for the R8-mannose/PEI/mcDNA-E7 system, it is clear that maintaining the N/P ratios of PEI and varying the N/P ratios of R8-mannose or by maintaining only the N/P ratios of R8-mannose and increasing the N/P ratios of PEI, there is an increment of the surface charge. The proportion of primary amines from the polymer in relation to phosphate groups from the plasmid results in an increase of the positive charge on the systems surface. The systems that proved to be the most suitable for delivery purposes based on size and surface charge were those prepared at an R8-mannose N/P ratio of 2 and PEI N/P ratio of 10, because they exhibit the smallest size (234.56 nm) and the highest zeta potential (+14.67). The formulation of mcDNA systems conjugated with PEI and R8-mannose results in the formation of ternary systems with higher positive zeta potential values when compared with R8-mannose binary systems. Some studies have reported the formation of complexes additionally packaged with PEI to construct ternary systems, which showed resistant properties against serum proteins and rapid cellular uptake, leading to improved gene transfer efficiency [41]. Song and collaborators prepared a novel polyethyleneimine-RRRRRRRR(R8)-heparin (HPR) nanogel as an efficient gene delivery system. The R8 peptide was grafted onto PEI (R8-PEI) to increase the charge density, reduce the toxicity of the gene delivery system, and thus enhance cellular uptake and gene transfection efficiency [42]. Our results from DLS measurements, showing that the addition of PEI to the R8-mannose/mcDNA-E7 system not only considerably reduces the size of particles but also increases the positive charge they carry, agree well with the results found in the literature.

### 3.4. Stability Assay

To evaluate the stability of both systems in the extracellular compartment and the protection that the systems confer to the DNA vector encoding HPV E7 gene, formulations at various N/P ratios were incubated at 0, 1, and 4 h with DMEM medium supplemented with 10% FBS and with a trypsin solution at 37 °C. Figure 6 shows the obtained results.

Data obtained for R8-mannose/mcDNA-E7 systems show the presence of mcDNA in the supernatants of all formulations conceived at different N/P ratios at 0 h of incubation with the complete medium (Figure 6A). The electrophoretic profile also suggests a partial degradation of mcDNA in some cases. The same formulations were incubated with trypsin (Figure 6A) in another experiment to evaluate its action regarding the protection of the mcDNA-E7 vector. Once more, it was shown that the system is not able to maintain its integrity when incubated with trypsin. The decomplexation of mcDNA occurs at 0 h for the complete medium and trypsin incubations, suggesting the instability of these binary systems, which can compromise the protection, carriage, and in vitro and in vivo cell transfection efficiency [43]. On the other hand, the data obtained for R8-mannose/PEI/mcDNA-E7 formulations at different R8-mannose N/P ratios and both PEI N/P ratios (5 and 10) incubated with complete medium and trypsin at different times indicated the higher stability of ternary systems (Figure 6B,C). The electrophoretic migration of supernatants resulting from these experiments reveal absence of mcDNA in all incubations made with trypsin (Figure 6B,C) and a vestigial presence in incubation experiments with complete medium, mainly after 4 h period of incubation (Figure 6B,C). Curiously, this behavior is more evident in formulations with a PEI N/P ratio of 10. Nevertheless, and as discussed before, PEI has a higher capacity to condense DNA as it interacts strongly with this molecule. Therefore, these data suggest that R8-mannose/PEI/mcDNA-E7 ternary systems are more suitable than R8-mannose/mcDNA-E7 binary systems for delivery of DNA vaccines, probably due to the presence of PEI [42].

### 3.5. Biocompatibility Evaluation

The biocompatibility of the developed systems was evaluated through resazurin assay on human fibroblast and Raw 264.7 cells to determine if the studied systems had any toxic effect towards the cells once transfected. The results on human fibroblast and Raw 264.7 cells at 24 and 48 h for the various carriers at different N/P ratios are summarized in Figure 7.

It was observed that none of particles are toxic to the Human fibroblast cells since cellular viability shows values superior to 80% regardless the transfection period. The results on Raw 264.7 cells are identical once all systems have a viability greater than 80% even 48 h after transfection. Branched PEI polymers are known to compact DNA more efficiently due to the higher density of its primary amine groups. However, the cytotoxic effect associated with branched polymer structures must be taken into consideration, especially with branched PEI 25 kDa since it can induce cell membrane damage and initiate apoptosis. As found before, the molecular weight, the architecture of the polymer and the N/P ratio are all relevant parameters that can influence cellular viability. The N/P ratio appears to be the main factor determining cellular viability as branched PEI polymers are only biocompatible for the lower N/P ratios. The higher the N/P ratio used at vector preparation step, the higher the cytotoxicity of PEI 25 kDa. Nevertheless, it does not compromise its use, at least, until N/P ratio 15. To reduce the cellular toxic effects of branched PEI 25 kDa, lower N/P ratios should be considered so as not to increase the content of free amines that are responsible for cytotoxicity [19]. Considering that some studies report percentages of cell viability above 80% are considered non-cytotoxic [44], these results confirm an improvement in biocompatibility of PEI-based vectors, probably due to the presence of R8 peptide on the formulations.

### 3.6. Cellular Uptake and Intracellular Location of Complexes

The capacity for cellular uptake and internalization of the developed systems was evaluated. The uptake of PEI/mcDNA and R8-mannose/PEI/mcDNA systems into Raw 264.7 cells and their intracellular co-localization after 3 h of transfection was visualized by fluorescence confocal microscopy. The images are presented in Figure 8.

Nuclei were stained blue by DAPI, while green represented the mcDNA stained with FITC. The cell live imaging presented in Figure 8A corresponding to non-transfected cells (control) shows that cells do not exhibit green fluorescence signals. According to Figure 8B, corresponding to PEI/mcDNA formulations in the absence of R8-mannose, it is clear that mcDNA systems have less ability to reach the nuclei of target cells. Figure 8C, corresponding to the transfection mediated by R8-mannose/PEI/mcDNA systems, shows the presence of stained mcDNA into the cells, revealing higher ability from these systems to reach the nuclei of target cells. Therefore, the results suggest that these systems are able to overcome both extracellular and intracellular barriers and the presence of R8-mannose seems to efficiently improve the cell entry, internalization, and mcDNA accumulation into the nucleus [15,45]. Once inside the nucleus, it is expected that transcription and expression of the target gene will occur.

### 3.7. Expression of E7 Gene

The cellular transfection mediated by PEI/mcDNA systems was monitored in the absence and presence of R8-mannose to evaluate the influence on the systems this process, namely on E7 gene expression. The characteristics exhibited by the PEI/mcDNA complexes are strongly dependent on the N/P ratio used. A higher N/P ratio leads to a strong polymer amine density that can efficiently condense the mcDNA molecule and form delivery systems with a higher positive charge, favoring the interaction with negatively charged proteoglycans present in the cell membrane and facilitating their entry into the cell [46].

RT-PCR was used to evaluate the transcription efficiency of the E7 gene into the cells transfected by PEI/mcDNA and R8-mannose/PEI/mcDNA systems. Non-transfected cells were used as control and amplification of E7 transcripts was performed using specific primers. Samples were then analyzed using 1% agarose gel electrophoresis, the results of which are presented in Appendix A. However, as the assessment of the band intensities was in some cases unclear, the RT-qPCR technique was employed to quantify E7 expression levels, as this is a more precise method allowing for an accurate evaluation. The obtained results for gene expression after transfection with the developed systems are presented in Figure 9.

As shown in Figure 9, there is a general increase in the expression of E7 transcripts in relation to non-transfected cells (control). In the RAW 264.7 cell line, R8-mannose/PEI/mcDNA ternary systems show higher levels of E7 transcripts when compared to PEI/mcDNA binary systems. These results are in agreement with the previous tendency observed in confocal experiments (Figure 8), suggesting that the presence of R8-mannose influences the systems’ internalization, therefore favoring the subsequent processes of nucleus targeting and gene expression. In fact, this behavior could be related to the presence of the R8 peptide, which has shown quite satisfactory results in terms of cell membrane uptake and nuclear localization [14,16]. R8-mannose/PEI/mcDNA ternary systems at a PEI N/P ratio of 5 have higher levels of gene transcription than ternary systems at a PEI N/P ratio of 10. These results indicate that the PEI N/P ratio of 10 can reduce the proportion of R8-mannose included in the ternary system (probably due to the charge repulsion), decreasing the positive effects of this mannose-R8 conjugate in terms of cell recognition and internalization, consequently affecting the transcription of the E7 gene.

As the PEI/mcDNA N/P ratio of 5:1 and R8-mannose/PEI/mcDNA N/P ratio of 2:5:1 showed successful results in Raw 264.7 cells, they were chosen in an attempt to understand and clarify the influence of R8-mannose in human fibroblast cells (Figure 9B). The obtained results followed the same tendency as for RAW 264.7 cells, since ternary systems showed higher levels of E7 transcripts than binary systems, suggesting a positive influence of R8 in cell uptake and nucleus accumulation [15,31]. However, comparing the E7 gene transcription obtained by the R8-mannose/PEI/mcDNA ternary system in both cell lines, higher levels were attained in Raw 264.7 cells than in human fibroblast cells. These results could be related to the cumulative effects of mannose ligands, which can improve the system internalization through the recognition of mannose receptors of RAW cells. It is well known that Raw 264.7 cells expressing moderate mannose receptors might constitute a suitable and valuable opportunity to investigate the transfection efficiency of mannosylated systems [24,45]. Some studies have reported applications of mannosylated systems with improved transfection efficiency, whose results confirm that mannosylated systems are more effective for gene transcription and expression in Raw 264.7 cells [24]. Once again, the results suggest that the presence of both mannose and R8 may influence the extent of cellular internalization of the systems, preferentially by APCs such as macrophages. This fact in turn may dictate the success of gene transcription and consequently protein expression, and therefore the therapeutic effect.

A successful anti-cancer vaccine depends on its ability to induce humoral and cellular immunity against a specific antigen. Antigens are known to be delivered by various dendritic cell receptors, including C-type lectin receptors (CLRs), which are important pattern recognition receptors involved in the induction of adaptive immunity against pathogens [47,48]. Numerous receptors expressed on DCs have been identified, and each of them have shown potential as targets for cancer vaccine design. Most studies to date have used mannose to target the mannose receptors, as it is highly expressed on DCs and macrophages and plays a key role in antigen recognition [45,48]. However, some studies have shown good results with other CLRs, such as langerin, DC-SIGN, and others. A comparative study between DC-SIGN and langerin showed functional differences, despite similarities in carbohydrate recognition domains. As with mannose receptors, DC -SIGN also recognizes carbohydrates on pathogens mediating endocytosis, thereby activating the adaptive immune response against pathogens [47,48,49]. Although our data show evidence of efficient E7 gene transfection via the developed R8 mannose/PEI/mcDNA-E7 systems, further studies are needed to evaluate the immune response. It would also be relevant to verify whether these nanoparticles, when recognized by pathogen recognition receptors such as C-type lectin, could stimulate the immune response, and thus serve as vaccine adjuvants.

## 4. Conclusions

The main challenge associated with DNA vaccines and its main limitation for clinical application is the delivery barriers to targeted immune cells, which obstruct the stimulation of effective antigen-specific immune responses in humans, which ultimately, leads to low therapeutic efficacy. In this report, novel delivery systems have been developed, aiming to add a significant contribution to DNA vaccines by exploring the assets of non-viral systems. Pursuing this goal, R8-mannose/mcDNA-E7 binary and R8-mannose/PEI/mcDNA-E7 ternary systems have been conceived at several N/P ratios and their physicochemical properties have been assessed. The carriers have shown a set of properties related to their morphology, size, surface charge, and cytotoxic profile, which opens the possibility to use them as suitable delivery vehicles. In vitro studies and a comparison between the formulated carriers revealed that R8-mannose/PEI/mcDNA-E7 systems are more functional DNA vaccine delivery carriers. These ternary systems, either incubated with trypsin or DMEM medium, were able to efficiently condense mcDNA, since there was no evidence of decomplexation or degradation of particles. The addition of the PEI to the R8-mannose/mcDNA-E7 system not only considerably reduces the size of the particles but also increases the zeta potential; the vectors were also revealed to be biocompatible. Finally, expression of the E7 gene was more intense with ternary systems, which suggests the influence of mannose and R8 for receptor recognition and cellular internalization. To guarantee a better effect of R8-mannose/PEI in the recognition of mannose receptors and internalization by R8, control of the proportion of PEI is needed.

This work constitutes a significant advance in the conception of non-viral delivery systems to carry DNA vaccines, offering to this field a suitable tool to be further evaluated.

## Figures and Tables

**Figure 1 pharmaceutics-13-00673-f001:**
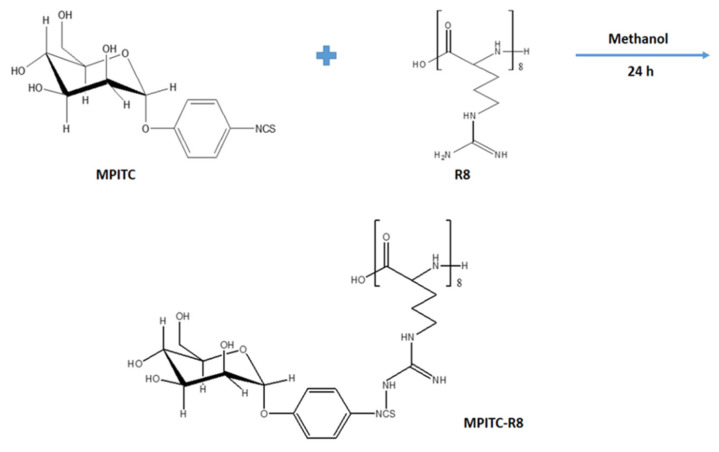
Synthesis scheme of MPITC-R8 conjugate.

**Figure 2 pharmaceutics-13-00673-f002:**
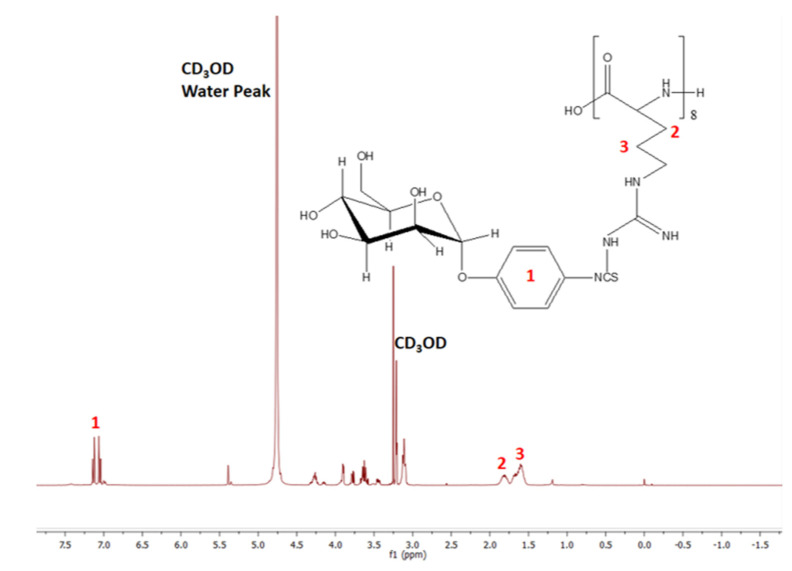
^1^H NMR spectrum of MPITC-R8 conjugate.

**Figure 3 pharmaceutics-13-00673-f003:**
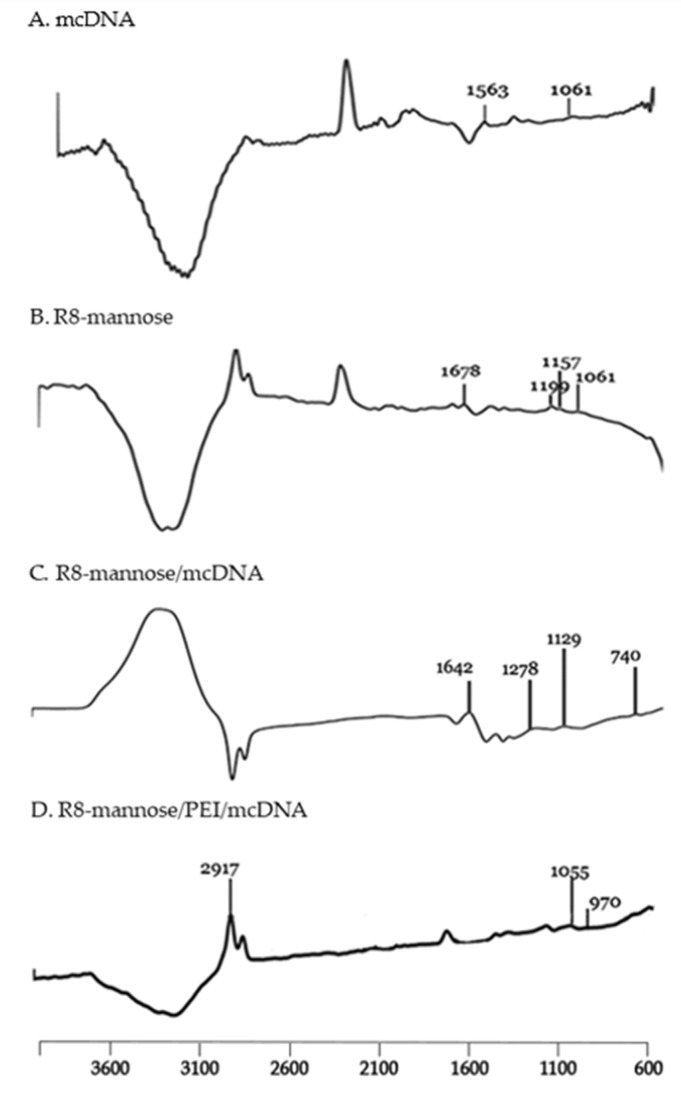
FTIR spectra (absorbance *versus* wavenumber) of mcDNA (**A**), R8-mannose (**B**), R8-mannose/mcDNA (**C**), and R8-mannose/PEI/mcDNA (**D**) samples.

**Figure 4 pharmaceutics-13-00673-f004:**
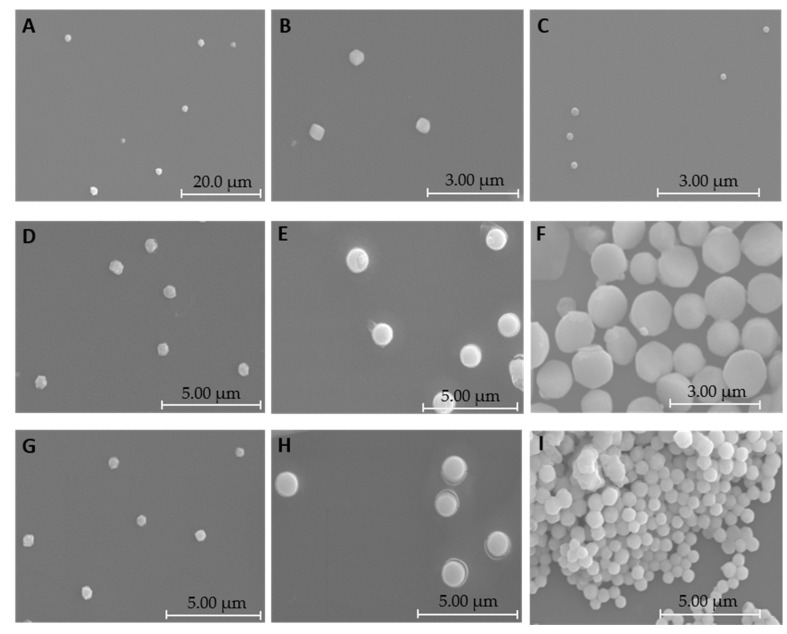
Scanning electron micrographs of particles formulated at the (**A**) R8-mannose/mcDNA N/P ratio of 1:1; (**B**) R8-mannose/mcDNA N/P ratio of 1.5:1; (**C**) R8-mannose/mcDNA N/P ratio of 2:1; (**D**) R8-mannose/PEI/mcDNA N/P ratio of 1:5:1; (**E**) R8-mannose/PEI/mcDNA N/P ratio of 1.5:5:1; (**F**) R8-mannose/PEI/mcDNA N/P ratio of 2:5:1; (**G**) R8-mannose/PEI/mcDNA N/P ratio of 1:10:1; (**H**) R8-mannose/PEI/mcDNA N/P ratio of 1.5:10:1; and (**I**) R8-mannose/PEI/mcDNA N/P ratio of 2:10:1.

**Figure 5 pharmaceutics-13-00673-f005:**
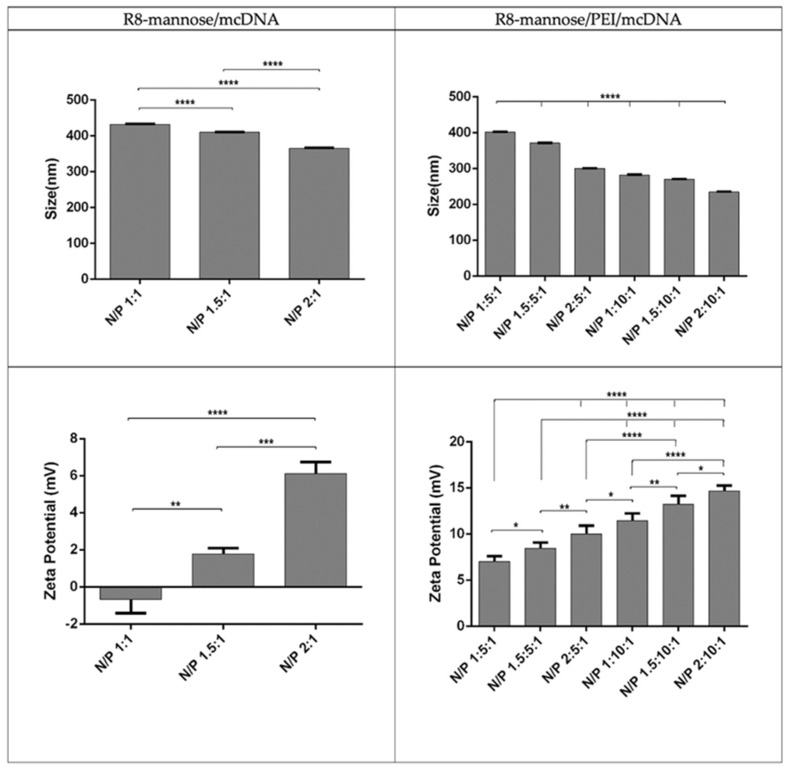
Mean sizes (nm) of binary and ternary systems and average zeta potentials (mV) of binary and ternary systems formulated at different N/P ratios. The values were calculated with the data obtained from three independent measurements (mean ± S.D., *n* = 3); * *p* < 0.05; ** *p* < 0.01; *** *p* < 0.001; **** *p* < 0.0001.

**Figure 6 pharmaceutics-13-00673-f006:**
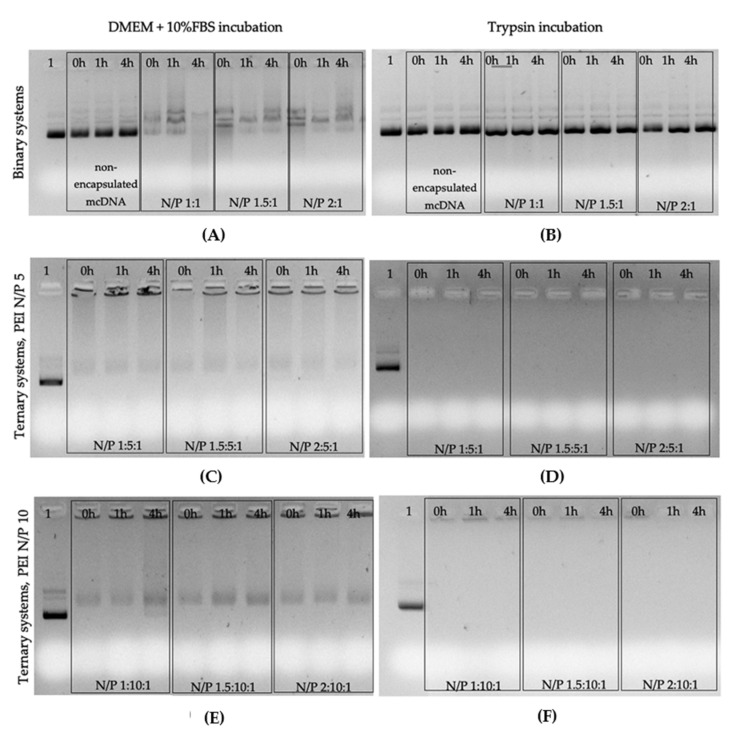
Electrophoretic analysis of mcDNA formulations after incubation with DMEM + 10% FBS (**A**,**C**,**E**) and with trypsin (**B**,**D**,**F**): 1- mcDNA control; A and B: R8-mannose/mcDNA-E7; C and D: R8-mannose/PEI/mcDNA-E7 systems formulated by maintaining a PEI N/P ratio of 5 and changing R8 N/P ratios; E and F: R8-mannose /PEI/mcDNA-E7 systems formulated by maintaining PEI N/P ratio of 10 and changing R8 N/P ratios.

**Figure 7 pharmaceutics-13-00673-f007:**
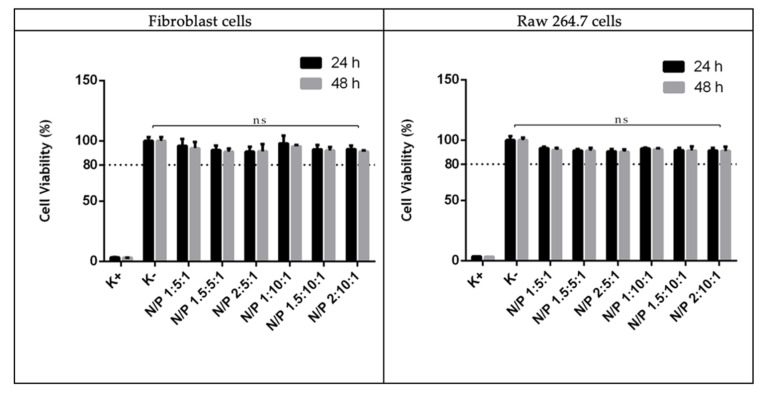
Cellular viability of human fibroblast and Raw 264.7 cells after 24 and 48 h incubation with R8-mannose/PEI/mcDNA particles at various N/P ratios. Percent viability is expressed relative to negative control cells. Non-transfected cells were used as negative control. Data obtained from three independent measurements (mean ± S.D., *n* = 3).

**Figure 8 pharmaceutics-13-00673-f008:**
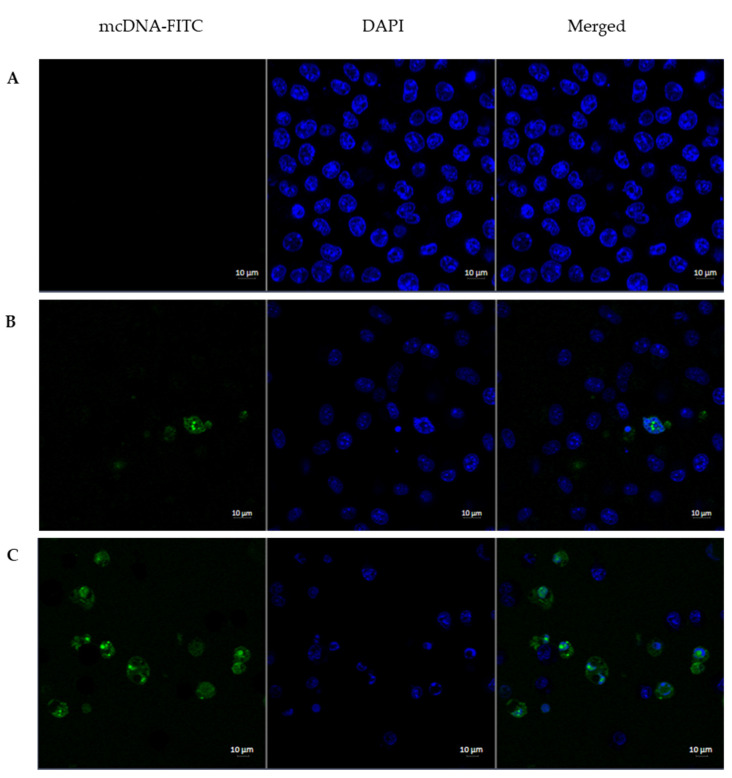
The transfection ability and intracellular co-localization of PEI/mcDNA (N/P ratio of 5:1) and R8-mannose/PEI/mcDNA (N/P ratio of 2:5:1) systems was investigated by fluorescence confocal microscopy. Nuclei were stained blue by DAPI, while green represents the mcDNA stained with FITC. Live cell images: Raw 264.7 non-transfected cells (**A**); Raw 264.7 cells after 3 h of transfection with PEI/mcDNA (N/P ratio of 5:1) system (**B**) and Raw 264.7 cells after 3 h of transfection with R8-mannose/PEI/mcDNA (N/P ratio of 2:5:1) system (**C**).

**Figure 9 pharmaceutics-13-00673-f009:**
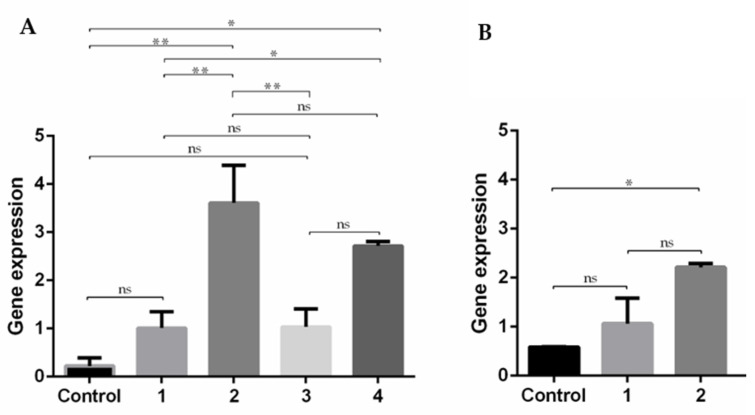
RT-qPCR of E7 expression levels in RAW 264.7 cells (**A**) and human fibroblast cells (**B**): Control–non-transfected cells; 1- PEI/mcDNA N/P ratio 5:1; 2- R8-mannose/PEI/mcDNA N/P ratio 2:5:1; 3- PEI/mcDNA N/P ratio 10:1; 4- R8-mannose/PEI/mcDNA N/P ratio 2:10:1. Data obtained from three independent measurements (mean ± S.D., *n* = 3); * *p* < 0.05; ** *p* < 0.01: ns—no significance.

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
