# Peer review of "Synthesis and Characterization of Mannosylated Formulations to Deliver a Minicircle DNA Vaccine"

_pharmaceutics, 2021, doi:10.3390/pharmaceutics13050673_

Round 1

Reviewer 1 Report

In the paper " Synthesis and characterization of mannosylated formulations to deliver a minicircle DNA vaccine ", the authors describe the formulation of binaries and ternary delivery systems for DNA-based vaccines, comparing their characteristics in terms of morphology, size, Z potential, toxicity, internalization capacity and transcription of the carried transgene.

The work is of interest and can be useful in the design and development of new non-viral DNA delivery systems in the field of DNA-based vaccine.

However, the work shows a clearer and more detailed first part describing the construction and the chemical characterization of the different systems, and a second part that needs some improvements. The English language should also be improved, with a better choice of some terms.

Comments:

-The methods describe the use of 1 ug of mcDNA for complexation in the various vehicle systems. Is it possible to measure the efficacy of complexation (for example by qRT-PCR) to know if all the formulated systems carry the same amount of DNA? Otherwise, the different vehicle efficiency may be due to the different amounts of DNA transducted.

-In the transfection and biocompatibility experiments, how many particles (and consequently, how much DNA) were used? These informations are missing. Have you seen any dose-dependent effect?

-Could the different transduction efficiency of RAW cells compared to fibroblasts be due to a different expression of the mannose receptor between the two cell lines? In other words, is it possible to analyze the different contributions of the  R8 and mannose in particle uptake? and/or to show the mannose receptor expression levels in both cell lines?

-In the confocal microscopy assay, why did you only use the 2:10:1 formulation? You should also add the data about the transfection using all the described formulations ( or at least the ones used in fig9-10), to better compare the different transfection efficacies of the various particles. In addition, PEI/mcDNA formulation in the absence of R8-mannose should be used as an appropriate control, to better demonstrate the contribution of the R8 peptide and the mannose to cell entry.

-As also reported in the text, RT-qPCR is a more precise method to evaluate E7 transcription levels. You should remove ( or include them in supplementary) non-quantitative RT-PCR data (Figure 9), as the assessment of the band intensities is in some cases unclear unless supported by densitometry data. Furthermore, the conclusions are already supported by the data in Figure 10. My suggestion is to include part of the fig.9 discussion to the description of the fig.10 data.

-Lines 797 to 799. The authors stated that the systems developed at  PEI N/P ratio 5 present more intense bands than PEI N/P ratio10, which may correspond to a higher cellular internalization. This last sentence should be removed unless supported by confocal microscopy studies ( see comment 5)

- The authors said: "Both systems with a PEI N/P 5 ratio have higher levels than vectors prepared with the N/P 10 ratio, which is in line with rt-PCR results”. But in Figure 10A the value shown by sample number 1 (ratio 5:1) looks the same as the value for the number 3 (ratio 10:1). There's maybe a mistake in the figure.

Minor comments:

The images in Figure 4 seem to use a different scale bar for each magnification. The scale bar used in each image should be clearly reported.

Fig 10 legend: lane2  and lane 4 are both described as “R8-mannose/PEI/mcDNA N/P ratio 2:5:1”.  Lane 4 should be ratio 2:10:1

Fig 10A: the p-value for the comparison among 3 and 4  and 2 and 4 should be added. Are these differences statistically significant?

Fig 10B: the p-value for the comparison among 1 and 2  should be added. Again, is this difference statistically significant?

Some words should be reviewed throughout the text. For example ( but not only) :

Line 44: Please, replace with “Among the different DNA vaccines,..”

Line 62: It should be:” between others”

Line 73: Please, correct “ nucleus accumulation in the nucleus”

Line 81: “PEI based DNA carriers “instead of “DNA based carriers of PEI”

Line 99: it should be: “In this work, binary and ternary delivery systems based on R8 peptide functionalized with mannose ligands with and without the PEI polymer were designed…”

Line 105: Please, replace with “expression of the gene encoded by the mcDNA vaccine carried by the developed systems”

Line 132: it should be: “production under induction conditions”

Line 411:” on DC” instead of “in DC”

Line798 “ to what may correspond a higher” should be “which may correspond to a higher”

Line 836: it should be” … have higher transcription levels”

Line 814 E7 transcription instead of E7 expression

Line 858: "assessed" instead of " researched"

Author Response

The authors would like to acknowledge the careful revision and pertinent reviewers’ comments and the possibility to improve our manuscript. All the questions were answered in the attached document and the recommended modifications were made, being properly highlighted at yellow in the revised manuscript file. 

Reviewer 2 Report

The manuscript of Serra et al shows results about the synthesis and characterization of mannosylated formulations that can be potentially used as a DNA vaccine delivery.  The authors struck a balance by focusing attention on their results and some references to previous results.

I just have some minor suggestions

By linking the results with the discussion, a greater emphasis was placed on the results but the discussion was weak.
There were aspects that could have been analyzed. For example, the authors only emphasize the property as DNA delivery, however, it is possible that these particles, when recognized by pathogen recognition receptors such as C-type lectin (eg mannose receptor, DC-SIGN, Langerin, etc), could stimulate the immune response and also serve as a vaccine adjuvant.
The demonstration of this effect was not evaluated in this work but could have been discussed as a potential effect that needs to be evaluated in future studies.
I suggest that results and discussion be separated to analyze those aspects that are of great relevance, even though they were not studied here, and also to delve into more results obtained by other authors linked to these findings. In this way, a better integrative analysis can be made, allowing to highlight the strengths of the work and what still needs to be evaluated.

Author Response

The authors would like to acknowledge the careful revision and pertinent reviewers’ comments and the possibility to improve our manuscript. All the questions were answered and the recommended modifications were made, being properly highlighted at yellow in the revised manuscript file.

Comments to the Author

The manuscript of Serra et al shows results about the synthesis and characterization of mannosylated formulations that can be potentially used as a DNA vaccine delivery.  The authors struck a balance by focusing attention on their results and some references to previous results. I just have some minor suggestions.

  1. By linking the results with the discussion, a greater emphasis was placed on the results but the discussion was weak. There were aspects that could have been analyzed. For example, the authors only emphasize the property as DNA delivery, however, it is possible that these particles, when recognized by pathogen recognition receptors such as C-type lectin (eg mannose receptor, DC-SIGN, Langerin, etc), could stimulate the immune response and also serve as a vaccine adjuvant.
    The demonstration of this effect was not evaluated in this work but could have been discussed as a potential effect that needs to be evaluated in future studies. I suggest that results and discussion be separated to analyze those aspects that are of great relevance, even though they were not studied here, and also to delve into more results obtained by other authors linked to these findings. In this way, a better integrative analysis can be made, allowing to highlight the strengths of the work and what still needs to be evaluated.

Response: We thank the reviewer for this comment, which gives us the possibility to improve the manuscript. We do recognize the potential value and significance of the reviewer´s suggestions and following his/her guideline, a more detailed analysis and discussion was included in the current version of the manuscript, allowing to highlight the strengths of the work and what still needs to be evaluated in future studies. Specially, we emphasized the DNA particles when recognized by pathogen recognition receptors such as C-type lectin (eg mannose receptor, DC-SIGN, Langerin, etc), could stimulate the immune response and also serve as a vaccine adjuvant. In this way, the additional information was considered:

“A successful anti-cancer vaccine depends on its ability to induce humoral and cellular immunity against a specific antigen. Antigens are known to be delivered by various dendritic cell receptors, including C-type lectin receptors (CLRs), which are important pattern recognition receptors involved in the induction of adaptive immun-ity against pathogens [1,2]. Numerous receptors expressed on DCs have been iden-tified, and each of them showed potential as a target for cancer vaccine design. Most studies to date have used mannose to target the mannose receptors, as they are highly expressed on DCs and macrophages, and play a key role in antigen recognition [2,3]. However, some studies have shown good results with other CLRs, such as langerin, DC-SIGN and others. A comparative study between DC-SIGN and Langerin showed functional differences, despite similarities in carbohydrate recognition domains. Like mannose receptors, DC -SIGN also recognizes carbohydrates on pathogens mediating endocytosis, thereby activating the adaptive immune response against pathogens [1,2,4]. Although our data show evidence of efficient E7 gene transfection by the de-veloped R8 mannose/PEI/mcDNA-E7 systems, further studies are needed to evaluate the immune response. It would also be relevant to verify whether these nanoparticles, when recognized by pathogen recognition receptors such as C-type lectin, could stim-ulate the immune response and thus serve as vaccine adjuvants.” Overall, the discussion improvement and the added text can be found on the Results and Discussion section, topic 3.7, “Expression of E7 gene” (please see page 16, 17 and 18).

Regarding the separation of Results and discussion, we also deeply appreciate this suggestion and we will, for sure, consider follow it in future manuscript written preparation to better analyze some aspects that are of great relevance and give an integrative analysis, highlighting the strengths of the work and what still needs to be evaluated. However, in the present work, we consider to maintain this format because at some points we fell the need to explain and discuss the results attained to justify the reasons of the following experiments. For instance, due to the instability of R8-mannose/mcDNA binary systems observed in the topic 3.4, Stability assay (pages 13 and 14), these systems are unable to maintain mcDNA integrity, compromising protection and thus transport and transfection efficiency. By this reason, the following experiments from this point will consider by using the PEI/mcDNA binary instead of R8-mannose/mcDNA binary system. We strongly hope, this time, the reviewer can accept this format.

  1. Conniot, J.; Scomparin, A.; Peres, C.; Yeini, E.; Pozzi, S.; Matos, A.I.; Kleiner, R.; Moura, L.I.F.; Zupančič, E.; Viana, A.S.; Doron, H.; Gois, P.M.P.; Erez, N.; Jung, S.; Satchi-Fainaro, R.; Florindo, H.F. Immunization with mannosylated nanovaccines and inhibition of the immune-suppressing microenvironment sensitizes melanoma to immune checkpoint modulators. Nat Nanotechnol 2019, 14, 891-901.
  2. Apostolopoulos, V.; Thalhammer, T.; Tzakos, A.G.; Stojanovska, L. Targeting antigens to dendritic cell receptors for vaccine development. J Drug Deliv 2013, 2013, 869718.
  3. Hossain, M.K.; Wall, K.A. Use of Dendritic Cell Receptors as Targets for Enhancing Anti-Cancer Immune Responses. Cancers (Basel) 2019, 11.
  4. Tang, C.K.; Sheng, K.C.; Apostolopoulos, V.; Pietersz, G.A. Protein/peptide and DNA vaccine delivery by targeting C-type lectin receptors. Expert Rev Vaccines 2008, 7, 1005-1018.

Round 2

Reviewer 1 Report

The authors have adequately addressed my comments, and the text has now improved.

Two minor errors I've spotted:

Line 574: “Moreover, the images indicated that the internalized vectors localize into the nucleus.”  Due to the text modification, this sentence is now rendundant, and it should be removed

Line 583:  “PEI/mcDNA systems have already shown satisfactory results concerning the uptake and internalization in cells. “In the figure 8 is shown that PEI / mcDNA have less ability to be internalized by target cells than R8-mannose system. So this sentence seems contradictory and should be removed.

Author Response

The authors would like to acknowledge the careful revision and pertinent reviewers’ comments and the possibility to improve our manuscript. All the questions were answered and the recommended modifications were made, being properly highlighted using the “Track Changes” function in Microsoft Word in the revised manuscript file.

The authors have adequately addressed my comments, and the text has now improved.

Two minor errors I've spotted:

Line 574: “Moreover, the images indicated that the internalized vectors localize into the nucleus.”  Due to the text modification, this sentence is now rendundant, and it should be removed

Response: We thank the reviewer for this comment, which gives us the possibility to improve the manuscript. The sentence was removed in order to avoid redundant information, please see line 574.

Line 583:  “PEI/mcDNA systems have already shown satisfactory results concerning the uptake and internalization in cells. “In the figure 8 is shown that PEI / mcDNA have less ability to be internalized by target cells than R8-mannose system. So this sentence seems contradictory and should be removed.

Response: We thank the reviewer for this comment, which gives us the possibility to improve the manuscript. The sentence was removed, please see line 582.